# Combined Evaluation of FGF23, Klotho, Myostatin, IL-6, and IL-10 as Potential Biomarkers in Monitoring Stable Renal Transplant Recipients

**DOI:** 10.3390/jcm14228131

**Published:** 2025-11-17

**Authors:** Öznur Kal, Sevsen Kulaksızoğlu, Oğuzhan Kahraman, Demet Yavuz, Siren Sezer

**Affiliations:** 1Department of Nephrology, Faculty of Medicine, Baskent University, 42080 Konya, Türkiye; 2Department of Biochemistry, Faculty of Medicine, Baskent University, 42080 Konya, Türkiye; sevsenk@baskent.edu.tr; 3Department of Urology, Faculty of Medicine, Baskent University, 42080 Konya, Türkiye; okahraman_1989@hotmail.com; 4Department of Nephrology, Faculty of Medicine, Samsun University, 55080 Samsun, Türkiye; demetdolu@hotmail.com; 5Department of Nephrology, Faculty of Medicine, Baskent University, 06790 Ankara, Türkiye; sirensezer@hotmail.com

**Keywords:** renal transplantation, FGF23, Klotho, myostatin, IL-10, IL-6

## Abstract

**Background:** This study investigates the levels of fibroblast growth factor 23 (FGF23), Klotho, interleukin-6 (IL-6), interleukin-10 (IL-10), and myostatin (Mstn) in stable kidney transplant recipients undergoing triple immunosuppressive therapy. Chronic kidney disease (CKD) disrupts the balance of several key minerals and hormones, leading to complications such as vascular calcification and cardiac issues. We aim to identify potential biomarkers for monitoring kidney function post-transplantation and to understand the inflammatory response induced by renal transplantation. **Methods:** A total of 122 renal transplant patients on triple immunosuppression were included. Patients were categorized based on their calcineurin inhibitor usage and compared with 110 healthy individuals in the control group. Body mass index (BMI), serum FGF23, Klotho, IL-6, IL-10, Mstn, C-reactive protein, creatinine, calcium, phosphorus, and albumin levels were analyzed. **Results:** The study reveals that FGF23, Klotho, and Mstn levels are significantly lower in transplant patients than in the control group, while IL-6 levels show no significant difference. A decrease in IL-10 levels is observed in transplant patients, suggesting a role in post-transplant inflammation. Additionally, a positive correlation is found between BMI and serum Mstn and Klotho levels, but no correlation with other measured parameters. **Conclusions:** The findings suggest that serum levels of FGF23, Klotho, and Mstn, along with IL-10, could serve as indicators of kidney function and inflammation in kidney transplant recipients, potentially guiding post-transplant care and management.

## 1. Introduction

Chronic kidney disease (CKD) is associated with many disturbances in the homeostasis of calcium, phosphate, parathyroid hormone (PTH), calcitriol, fibroblast growth factor 23 (FGF23), and Klotho. Osteoblasts and osteocytes secrete FGF23, which is a phosphaturic hormone. It contributes to the maintenance of mineral metabolism. Besides its phosphaturic action, it also inhibits renal 1 α-hydroxylase activity, the rate-limiting enzyme in the calcitriol synthesis [1]. Consequently, it reduces intestinal phosphate absorption while increasing renal phosphate excretion. FGF23 activates FGF23 receptors to provide its biological effects in a Klotho-dependent way [2]. The single-pass transmembrane protein Klotho is expressed in the parathyroid glands, the brain’s choroid plexus, and the kidney’s distal and proximal renal tubules [3]. It functions as a coreceptor of FGF23 receptors.

With declining renal function, FGF23 levels are increased, and Klotho levels are decreased in CKD. These alterations in FGF23 and Klotho levels lead to vascular calcification, cardiac fibrosis, cardiac hypertrophy, and mortality in CKD [4]. Furthermore, CKD is linked to several metabolic disorders that impact energy, protein, and muscle metabolism. It was reported that oxidative stress, uremic toxins, less physical exercise, and chronic or acute inflammation could increase muscular atrophy by increasing myokines [5,6]. Myokines are cytokines synthesized by myocytes in skeletal muscle due to muscle contraction. Myostatin (Mstn), also known as growth development factor-8, belongs to the superfamily of transforming growth factor-β. Mature muscle cells secrete this myokine. It prevents muscle cell differentiation and proliferation. There are other factors that negatively regulate muscle mass in CKD besides Mstn. Interleukin-6 (IL-6) is a pro-inflammatory cytokine and also a potent myokine for skeletal muscles. IL-6 inhibits protein synthesis by Mstn overexpression during CKD [7].

The most successful treatment for end-stage renal disease is renal transplantation. However, it can result in a severe inflammatory response by inducing renal and plasma cytokine synthesis. The host’s reaction to kidney transplant is influenced by the cytokine levels. Acute phase protein synthesis, T cell proliferation, B cell differentiation, and immunoglobulin production are all impacted by IL-6. The primary mediator of the acute phase response due to an inflammatory condition is C-reactive protein (CRP). It is produced in connection with IL-6. Furthermore, IL-6 might have a role in fibrogenesis and endothelial function [8].

Additionally, the success or failure of kidney transplantation depends on the balance between pro-inflammatory and anti-inflammatory cytokines. The anti-inflammatory interleukin 10 (IL-10) inhibits macrophages and suppresses the Th-1 type immune response [9]. IL-10 has a role in controlling inflammation, in contrast to IL-6. IL-10 and IL-6 levels vary with the time after renal transplantation. It has previously been documented that serum FGF23 and Klotho levels can predict the course of CKD and the outcome of transplants [10,11].

Although patients are clinically stable after renal transplantation, renal impairment persists. Determination of potential biomarkers can be helpful in monitoring kidney function after renal transplantation. Despite several studies being conducted in recent years, there is still a debate about the best marker to better define inflammation and facilitate early detection. This study aimed to evaluate the levels of FGF23, Klotho, IL-6, IL-10, and Mstn in relation to subjects without comorbidities in stable kidney transplant recipients receiving triple immunosuppressive therapy.

## 2. Materials and Methods

This study included 122 renal transplant patients who were followed up in the Department of Nephrology, Baskent University Hospital, between 2021 and 2022. Every patient was on the triple immunosuppression, including prednisolone, mycophenolate mofetil, and calcineurin inhibitors (either tacrolimus, cyclosporine, or sirolimus). Based on their calcineurin inhibitor usage, the renal transplant patients were divided into three groups as follows: Group A (Cyclosporine), Group B (Tacrolimus), and Group C (Sirolimus). The control group consisted of 110 healthy volunteers. Our aim in dividing patients into three groups according to calcineurin inhibitor usage was to investigate whether different immunosuppressive regimens had distinct effects on biomarker levels. All samples were obtained during routine follow-up visits, and the mean post-transplant period was 6.3 ± 4.9 years. Patients with less than 6 months post-transplantation were excluded. Patients with a history of acute rejection and retransplant recipients were excluded. Only patients with stable graft function and no prior rejection episodes were included. Patients with acute infections, congestive heart failure, coronary artery disease, a history of surgery within the last 3 months, active malignancies, autoimmune diseases, thrombotic diseases, and any gastrointestinal diseases were excluded from the study. Of 134 screened transplant recipients, 12 were excluded due to active infection (n = 5), autoimmune disease (n = 4), cardiovascular comorbidity (n = 3). All blood samples were obtained at a single outpatient visit at least six months after transplantation during a stable follow-up period. Body compositions were analyzed with the BIA technique (BCM, Fresenius), estimating body mass index (BMI), muscle mass, and fat mass.

This study followed The Declaration of Helsinki’s Ethical Guidelines. The study was approved by Baskent University Institutional Review Board and Ethics Committee (Project no: KA16/226) and supported by Baskent University Research Fund. Every patient gave written and informed permission.

Blood samples were centrifuged at 3000 rpm for 10 min. Serum samples were kept at −80 °C until analysis. Serum creatinine, calcium, phosphorus, and albumin levels were measured by colorimetric-spectrophotometric method in an Abbott Architect C-8000 autoanalyzer (Abbott Laboratories, Abbott Park, IL, USA). Blood urea nitrogen (BUN) levels were measured by Abbott Architect C-8000 autoanalyzer (Abbott Laboratories, Abbott Park, IL, USA) using the enzymatic method. Estimated glomerular filtration rates (eGFR) were obtained from the CKD-EPI formula. Serum C-reactive protein (CRP) levels were measured by Abbott Architect C-8000 autoanalyzer (Abbott Laboratories, Abbott Park, IL, USA) using turbidimetric method. Serum PTH levels were measured by chemiluminescence microparticle immunoassay method in an Abbott Architect i 2000 autoanalyzer (Abbott Laboratories, Abbott Park, IL, USA). Hemoglobin levels were measured using Abbott CELL-DYN Ruby autoanalyzer (Abbott Laboratories, Abbott Park, IL, USA).

Serum FGF23 levels were measured by the enzyme-linked immunosorbent assay (ELISA) method using Sunredbio kit (Cat. No 201-12-0060, Shanghai Sunred Biological Technology Co. Ltd., Shanghai, China). The intra-assay precision, expressed as coefficient variation (CV), was <10%. The inter-assay precision, expressed as CV, was <12%. The concentrations of serum FGF23 were given in pg/mL. Serum Klotho levels were measured by ELISA method using Sunredbio kit (Cat. No 201-12-2782, Shanghai Sunred Biological Technology Co. Ltd., Shanghai, China). The intra-assay CV was <10%. The inter-assay CV was <12%. The concentrations of serum Klotho were given in ng/mL. Serum Mstn levels were measured by ELISA method using Sunredbio kit (Cat. No 201-12-0404, Shanghai Sunred Biological Technology Co. Ltd., Shanghai, China). The intra-assay CV was <8%. The inter-assay CV was <11%. The concentrations of serum Mstn were given in ng/L. The immunoenzymetric assay method was used to measure the levels of serum IL-6 and IL-10, using DIAsource kits (Cat. No KAP1261 and KAP1321, respectively, DIAsource ImmunoAssays S. A., Ottignies-Louvain-la-Neuve, Belgium). Serum IL-6 and IL-10 concentrations were given in pg/mL. The intra-assay CV of IL-6 was <4.2%. The inter-assay CV of IL-6 was <4.4%. The intra-assay CV of IL-10 was <3.7%. The inter-assay CV of IL-10 was <2.7%.

### Statistical Analysis

Every outcome was presented as mean ± standard deviation. Statistical analysis was performed by SPSS version 21.0 (IBM, Armonk, NY, USA). The Kolmogorov–Smirnov test was utilized to analyze the normality of the values. More than two parameters were compared using ANOVA test. Post hoc analysis was carried out using Tukey’s test. Categorical parameters were analyzed with chi-squared tests. Correlations between parameters were evaluated by the Pearson or Spearman correlation test. A p-value of less than 0.05 was considered to show a statistically significant result.

## 3. Results

The demographic and clinical characteristics of the patients are recorded in Table 1. Patients were categorized into three groups based on their calcineurin inhibitor usage. The patients in Group A (14 males, 22 females; mean age, 44.5 ± 11.14) used cyclosporine. The patients in Group B (19 males, 39 females; mean age, 38.59 ± 10.18) used tacrolimus. And the patients in Group C (9 males, 19 females; mean age, 40.1 ± 11.04) were subjected to sirolimus therapy. The mean post-transplant period was 6.3 ± 4.9 years. The control group consisted of 110 healthy volunteers (38 males, 72 females; mean age, 40.1 ± 11.04). There were no statistically significant differences between the groups regarding age and gender (*p* = 0.071, *p* = 0.89, respectively). BMI and muscle mass did not exhibit statistically significant levels among the four groups (*p* = 0.08, p = 0.22, respectively). But in contrast to the other groups, the patients in Group C had noticeably lower levels of fat mass (*p* = 0.038). Transplant patients had significantly increased blood creatinine and BUN levels (*p* < 0.001 and *p* = 0.013, respectively). When transplant patients were compared to the control group, a significant decline in eGFR was noted (*p* < 0.001). There was no discernible variation among the four groups regarding serum calcium, phosphorus, albumin, PTH, CRP, and hemoglobin levels (*p* = 0.91, *p* = 0.68, *p* = 0.69, *p* = 0.31, *p* = 0.65, *p* = 0.06, respectively).

FGF23, Klotho, and Mstn levels did not differ between the renal transplant subgroups (*p* = 0.314, *p* = 0.26, *p* = 0.156, respectively). FGF23, Mstn, and IL-10 were significantly lower in the renal transplant patients than in the control group (*p* < 0.001, *p* < 0.001, *p* = 0.01, respectively). IL-6 levels did not significantly differ between the renal transplant subgroups nor between the renal transplant patients and the control group. No statistical difference was observed between the renal transplant subgroups with respect to IL-10 levels (*p* = 0.627) (Table 2).

A statistically significant positive correlation was obtained between BMI and serum Mstn levels (r = 0.183, *p* = 0.034). And also, there was a positive correlation between BMI and serum klotho levels (r = 0.0183, *p* = 0.032). Nevertheless, no statistically significant correlation was seen between serum creatinine, BUN, eGFR, calcium, phosphorus, PTH, CRP levels, and serum FGF23, Klotho, Mstn, IL-6, and IL-10 levels.

Pearson’s correlation analysis showed significant positive correlations between myostatin and FGF23 (r = 0.582, *p* < 0.001) and between myostatin and Klotho (r = 0.702, *p* < 0.001), as well as a weaker but significant association between FGF23 and Klotho (r = 0.335, *p* < 0.001). No significant correlations were observed between IL-6 or IL-10 and the other parameters (*p* > 0.05) in transplant patients.

## 4. Discussion

Kidney functions resume after a successful kidney transplantation. Accumulated waste products and electrolytes are excreted. Regulatory hormones in bone and mineral metabolism attenuate. However, some abnormalities persist in patients. Phosphate retention in CKD patients frequently results in elevated levels of FGF23 prior to kidney transplantation. Increases in serum FGF23 and decreases in serum Klotho serve as early indicators of renal impairment. Barker and associates showed that serum Klotho levels declined in mild CKD patients and preceded the elevation of serum FGF23 [12]. Hu and colleagues found a correlation between the magnitude of decrease in urinary Klotho was correlated with the severity of decline in eGFR in CKD patients [13]. Following a successful kidney transplantation, urinary phosphate excretion is normally increased by the relatively high FGF23 levels, leading to renal phosphate wasting and low serum phosphate levels. Numerous investigations have demonstrated that values decrease following a successful renal transplantation [14,15]. Evenepoel and associates observed a 95% decline in FGF23 levels after 3 months post-transplantation [16]. Prasad and associates demonstrated that FGF23 levels progressively returned to normal in all transplant patients [17]. In our study, transplant patients had significantly increased blood creatinine and BUN levels (*p* = 0.013, *p* < 0.001, respectively). When transplant patients were compared to the control group, a significant decline in eGFR was noted (*p* < 0.001). There was no significant difference among the groups with respect to serum calcium, phosphorus, albumin, and PTH levels (*p* = 0.91, *p* = 0.68, *p* = 0.69, *p* = 0.31, respectively). FGF23 levels were significantly lower in the renal transplant patients than in the control group (*p* < 0.001). As renal functions normalize, FGF23 levels return to baseline after transplantation. Despite the return of renal functions, the number of intact nephrons is still less in association with reduced Klotho expression. The length of time required in the post-transplant phase to restore these parameters to their baseline is noteworthy. In our study, post-transplantation period was 6.32 ± 4.94 years.

CKD is associated with a wasting syndrome. As eGFR declines, there seems to be a reduction in muscle mass and strength. Muscle loss comes from a disequilibrium between insulin-like growth factor-1 (IGF-1) and Mstn [18]. Yasar and associates measured the highest level of Mstn in the hemodialysis group and the lowest in the transplantation group [19]. In a study by Gil and associates, there was a significant improvement in both respiratory and peripheral muscle strength after six months following transplantation compared to the pretransplant period [20]. Kopple and associates observed that muscle Mstn mRNA transcripts in patients on maintenance hemodialysis decreased during physical exercise [21]. Another cause of muscle wasting in CKD is low-grade inflammation. According to Kaizu and associates, high circulating IL-6 and CRP levels are associated with low muscle mass [22]. And also, renal transplantation can itself elicit an inflammatory reaction. Mota and associates observed a peak of IL-6 pro-inflammatory cytokine in patients with high creatinine levels [23]. Moreover, Dahle and associates suggested that IL-6 may be a potential inflammatory marker to detect early loss of renal graft function [24]. IL-6 serves a dual biological role, acting both as a pro-inflammatory cytokine that contributes to graft injury and systemic inflammation, and as a myokine involved in skeletal muscle metabolism. The potential suppressive effects of immunosuppressive agents, particularly calcineurin inhibitors and corticosteroids, on cytokine expression may explain the relatively unchanged IL-6 levels observed in our cohort.

In contrast to IL-6, IL-10 controls inflammation by limiting the immune response. Chen and associates experimentally confirmed this protective action of IL-10 [25]. They showed that transduction of the IL-10 gene enhanced renal function and improved allograft survival in a rat model of kidney allograft rejection. However, Alves and associates did not discover any significant differences when comparing the levels of IL-6, IL-10, and eGFR groups [26]. In our study, Mstn levels did not vary among the renal transplant patients (*p* = 0.156). However, the renal transplant patients’ Mstn levels were much lower compared to the control group (*p* < 0.001). This can be explained by the fact that renal functions recover after renal transplantation.

Furthermore, low levels of Mstn in renal transplant patients can be linked to decreased Mstn expression as a result of immunosuppressive therapy-induced suppression of inflammation. Since inflammation is suppressed due to immunosuppressive therapy, IL-6 levels did not significantly differ both within renal transplant patients and between the renal transplant patients and control group in our study. A significant decrease was observed in IL-10 levels when comparing the renal transplant patients with the control group (*p* = 0.01).

### Limitations

There are some limitations of the current study. The sample size was relatively small, and the design was cross-sectional without longitudinal follow-up, which limits the generalizability of the results. The unequal subgroup sizes and differences in age and gender distribution might have influenced the results, and therefore the findings should be interpreted cautiously. Another limitation is that cytokine and growth factor levels were measured at a single time point, precluding evaluation of longitudinal changes. This study lacks non-transplant CKD comparison group. And there was limited information on co-medication and exact immunosuppressive drug levels.

## 5. Conclusions

To date, the relationship between FGF23, Klotho, Mstn, and cytokines have all been studied independently in relation to renal transplant patients in several studies. Nevertheless, no study has compared all of these factors in the same patient group. In this study, we analyzed the association of these variables altogether. Our results showed that subclinical inflammation may exist even in the absence of detectable levels of active inflammation in the blood. So combined evaluation of these variables would offer a more comprehensive picture of the clinical monitoring of renal transplant patients. Since these findings are based on a small group of patients, further prospective longitudinal studies with larger and more balanced cohorts are required to validate these findings.

## Figures and Tables

**Table 1 jcm-14-08131-t001:** Demographic and clinical characteristics of the study groups.

	Group A (Cyclosporine) (n = 36)	Group B (Tacrolimus) (n = 58)	Group C (Sirolimus) (n = 28)	Control Group (n = 110)	*p*
**Age (years)**	44.5 ± 11.4	38.59 ± 10.18	39.29 ± 10.98	40.1 ± 11.04	0.071
**Male/Female**	14/22	19/39	9/19	38/72	0.89
**BMI (kg/m^2)^**	28.15 ± 6.39	26.67 ± 4.94	25.09 ± 3.45	26.11 ± 4.72	0.08
**Muscle Mass (kg)**	52.41 ± 9.07	52.93 ± 11.07	53.14 ± 9.87	49.76 ± 10.75	0.22
**Fat Mass (kg)**	21.19 ± 12.3	19.24 ± 8.7	14.89 ± 6.66	19.94 ± 8.06	0.038 *****
**BUN (mg/dL)**	21.89 ± 16.03	17.61 ± 8.07	21.36 ± 7.89	15.24 ± 12.2	0.013 *****
**Creatinine (mg/dL)**	1.38 ± 0.83	1.2 ± 0.38	1.48 ± 0.46	0.9 ± 0.37	<0.001 *****
**eGFR**	66.27 ± 20.68	76.52 ± 21.77	57.92 ± 18.31	98.37 ± 20.75	<0.001 *****
**Calcium (mg/dL)**	9.49 ± 0.55	9.47 ± 0.48	9.52 ± 0.49	9.44 ± 0.39	0.91
**Phosphorus (mg/dL)**	3.6 ± 0.8	3.5 ± 0.78	3.63 ± 0.88	3.6 ± 0.55	0.68
**Albumin (g/dL)**	4.24 ± 0.39	4.3 ± 0.38	4.2 ± 0.3	4.32 ± 0.31	0.69
**PTH (ng/L)**	138.27 ± 188.3	166.75 ± 192.43	107.98 ± 57.64	81.37 ± 99.25	0.31
**CRP (mg/L)**	4.98 ± 6.52	5.53 ± 11.13	5.54 ± 12.61	3.72 ± 5.23	0.65
**Hemoglobin (g/dl)**	13.16 ± 1.72	13.66 ± 1.92	12.72 ± 1.71	13.58 ± 1.51	0.06

BMI, body mass index; BUN, blood urea nitrogen; eGFR, estimated glomerular filtration rate; CRP, C-reactive protein. Post hoc Tukey HSD analysis revealed that fat mass, BUN, creatinine, and eGFR levels showed statistically significant differences among groups, with the Control Group (*p* < 0.05). * Significant values.

**Table 2 jcm-14-08131-t002:** Comparison between the renal transplant patients grouped according to calcineurin inhibitor usage and the control group.

	Group A (Cyclosporine) (n = 36)	Group B (Tacrolimus) (n = 58)	Group C (Sirolimus) (n = 28)	Control Group (n = 110)	*p*
**FGF23 (** **pg** **/mL)**	186.48 ± 227.68	229.103 ± 289.30	140.37 ± 194.94	523.07 ± 365.28	<0.001 *
**Klotho (ng/mL)**	7.83 ± 5.75	7.54 ± 6.26	7.38 ± 3.90	11.83 ± 1.8	0.79
**Mstn (ng/L)**	693.06 ± 497.55	883.5 ± 752.81	633.85 ± 405.48	1284.21 ± 938.88	<0.001 *
**IL-6 (pg/mL)**	32.71 ± 18.91	28.09 ± 2.41	30.52 ± 13.55	32.15 ± 18.85	0.41
**IL-10 (pg/mL)**	35 ± 7.3	40.10 ± 4.76	33.37 ± 4.25	46.54 ± 20.23	0.01 *

FGF23, fibroblast growth factor 23; Mstn, myostatin; IL-6, interleukin 6; IL-10, interleukin 10. Post hoc Tukey HSD analysis revealed that FGF23, myostatin, and IL-10 levels showed statistically significant differences among groups, with Control Group demonstrating higher mean values compared with the other groups (*p* < 0.05). No significant intergroup differences were observed for Klotho, IL-6, or IL-10 levels (*p* > 0.05). * Significant values.

## Data Availability

The data that support the findings of this study are available on request from the corresponding author. The data are not publicly available due to privacy or ethical restrictions.

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
