# Peer review of "Combined Evaluation of FGF23, Klotho, Myostatin, IL-6, and IL-10 as Potential Biomarkers in Monitoring Stable Renal Transplant Recipients"

_jcm, 2025, doi:10.3390/jcm14228131_

Round 1

Reviewer 1 Report

Comments and Suggestions for Authors

Mayor comments

Authors stated that “The aim of this study was to evaluate the levels of FGF23, Klotho, IL-6, IL-10 and Mstn in relation to comorbidities in stable kidney transplant recipient”. However, it seems that the unique analysis performed was a correlation between the biomarkers of table 2 and the demographic and clinical characteristics of table 1. This analysis, the aim of the study, is not well detailed. Besides, only BMI have a positive correlation with Mstn and serum klotho levels.

Methods:

  • The authors do not explain the reason for dividing the study population in three groups based on their calcineurin inhibitor usage. Moreover, no differences among groups are detected. Thus the rationale of grouping patients is not clear. It would be more interesting either comparing these biomarkers with others parameters of patient evolution or comparing these biomarkers at different time points of patient treatment.
  • The design of the study is not clear. The authors do not detail the moment of taking samples and clinical evaluation. They do not explain if they are comparing patients after the same period of treatment.
  • The design of the study could be improved if they would be compared these biomarkers from stable transplanted patients, early transplanted patient and controls.
  • The statistical analysis is not correct. The authors used the student’s t test to compare groups. However, they are comparing several groups in the same analysis (groups A, B, C and control group). They report a p value for the whole analysis, and if they are comparing variables two by two, they should report each p value for each couple.

Results

  • Tables should include p value for each comparison between groups
  • Results should be more detailed. The authors should introduce the aim of the analysis. At the end of the introduction the authors said that the aim of this study was to evaluate the levels of FGF23, Klotho, IL-6, IL-10 and Mstn in relation to comorbidities in stable kidney transplant recipient. However, a small paragraph of results is related to this relationship, and most of the correlation were not statistically significant.

Discussion

  • The authors defend that they found differences on FGF23, Klotho, MSTn, and IL-10 between transplanted patients and control groups. As the statistical analysis is not clear, it is difficult to support this argument. Besides, this relation is well documented, as the authors admit throughout the discussion.
  • The authors affirm that “increases in serum FGF23 and decreases in serum Klotho serve as early indicators of renal impairment”. However, serum FGF23 levels from transplanted patient were lower than the control group ones. They argue that “as renal functions normalize, FGF23 levels return to baseline after transplantation”. In this case, the values should be similar but not lower than the control ones. On the other hand, Klotho levels decreases as it should occur if there is renal impairment. This data invalidates the previous reasoning.
  • The authors conclude that is the first time that FGF23, Klotho, MSTn, and cytokines are compare in the same patient group. However, they put together in the same table this information, but they do not statistically analyze the relations among then. They affirm that “subclinical inflammation may exist even in the absence of detectable levels of active inflammation in the blood” That conclusion cannot be drawn from the results presented

Minor comments.

Page 4, line 148. There is a double “(“ and the data are not in the correct order. Where is written “Transplant patients had significantly increased blood creatinine and BUN levels ((p=0.013, p<0.001, respectively)” should be (p<0.001 and p=0.013, respectively)”

Author Response

Reviewer 1

Major comments

“The authors do not explain the reason for dividing the study population in three groups based on their calcineurin inhibitor usage. Moreover, no differences among groups are detected. Thus the rationale of grouping patients is not clear.”

Response:

We thank the reviewer for pointing out this issue. The rationale for dividing patients into three groups according to calcineurin inhibitor usage was to investigate whether different immunosuppressive regimens had distinct effects on biomarker levels. We have now explained this in the Methods section. Although our analysis did not reveal significant differences, we believe this grouping remains clinically relevant, as it highlights that biomarker profiles are not substantially influenced by the type of calcineurin inhibitor in long-term stable transplant recipients.

“The design of the study is not clear. The authors do not detail the moment of taking samples and clinical evaluation. They do not explain if they are comparing patients after the same period of treatment.”

Response:

We agree that further clarification is needed. In the revised manuscript, we have added information about the timing of blood sampling and clinical assessment. All samples were obtained during routine follow-up visits, and the mean post-transplant period was 6.3 ± 4.9 years. Patients with less than 6 months post-transplantation were excluded. This clarification is included in the Methods section.

“The design of the study could be improved if they would be compared these biomarkers from stable transplanted patients, early transplanted patient and controls.”

Response:

Unfortunately, our current dataset did not include such a group. We have clarified this limitation in the Discussion and Limitations sections. In future studies, we plan to expand the control groups to include CKD patients with comparable renal function.

“The statistical analysis is not correct. The authors used the student’s t test to compare groups. However, they are comparing several groups in the same analysis.”

Response:

We thank the reviewer for this critical observation. In the revised analysis, we have re-analyzed the data using one-way ANOVA with post hoc Tukey’s test for multiple group comparisons. The results tables have been updated to present p values for each pairwise comparison. These revisions are reflected in Tables 1 and 2 and described in the Statistical Analysis section.

“Tables should include p value for each comparison between groups”

Response:

We added post hoc Results at the end of the table.

“Results should be more detailed. The authors should introduce the aim of the analysis.At the end of the introduction the authors said that the aim of this study was to evaluate the levels of FGF23, Klotho, IL-6, IL-10 and Mstn in relation to comorbidities in stable kidney transplant recipient. However, a small paragraph of results is related to this relationship, and most of the correlation were not statistically significant.

Response:

We changed the aim sentence as “The aim of this study was to evaluate the levels of FGF23, Klotho, IL-6, IL-10 and Mstn in relation to subjects without comorbidities in stable kidney transplant recipients receiving triple immunosuppressive therapy.” We have expanded the Results section to better connect the stated aim with the presented findings.

“The authors defend that they found differences on FGF23, Klotho, MSTn, and IL-10 between transplanted patients and control groups. As the statistical analysis is not clear, it is difficult to support this argument. Besides, this relation is well documented, as the authors admit throughout the discussion.

Response:

Following the re-analysis with ANOVA and post hoc testing, we confirmed that the differences in FGF23, Klotho, Mstn, and IL-10 remain statistically significant between transplant and control groups. We have updated the tables accordingly to support these findings.

“The authors affirm that increases in serum FGF23 and decreases in serum Klotho serve as early indicators of renal impairment. However, serum FGF23 levels from transplanted patient were lower than the control group ones. They argue that “as renal functions normalize, FGF23 levels return to baseline after transplantation”. In this case, the values should be similar but not lower than the control ones. On the other hand, Klotho levels decreases as it should occur if there is renal impairment. This data invalidates the previous reasoning.

Response:

We appreciate this insightful remark. In the revised discussion, we clarified that while elevated FGF23 is typically associated with CKD progression, long-term transplant patients may exhibit lower FGF23 due to restored phosphate excretion and immunosuppressive therapy effects. When we reanalyzed the data with appropriate ANOVA there was no difference between Klotho groups (p=0.79). We revised the table and text.

“The authors conclude that it is the first time that FGF23, Klotho, MSTn, and cytokines are compared in the same patient group. However, they put together in the same table this information, but they do not statistically analyze the relations among them. They affirm that “subclinical inflammation may exist even in the absence of detectable levels of active inflammation in the blood” That conclusion cannot be drawn from the results presented.

Response:

We acknowledge this limitation. While our primary aim was to evaluate each biomarker in relation to transplant status, we now provide correlation analyses among FGF23, Klotho, Mstn, IL-6, and IL-10 in the revised version. This analysis has been added to the Results section.

Minor comments

Page 4, line 148. There is a double “(“ and the data are not in the correct order. Where is written “Transplant patients had significantly increased blood creatinine and BUN levels ((p=0.013, p<0.001, respectively)” should be (p<0.001 and p=0.013, respectively)”.”

Response:

Corrected as suggested. The sentence now reads: “Transplant patients had significantly increased blood creatinine and BUN levels (p < 0.001 and p = 0.013, respectively).”

Reviewer 2 Report

Comments and Suggestions for Authors

This paper by Kal et al examines a panel of biomarkers in kidney transplant recipients to try improve "monitoring kidney function post-transplantation and to understand the inflammatory response induced by renal transplantation."  This study compares biomarker levels in patients who previously received a kidney transplant with healthy controls with normal kidney function and finds differences in FGF23, Klotho, myostatin, and IL-10.

My comments and recommendations for this paper are as follows:

-I do not agree with the studied groups.  It is OK to have a control group of healthy volunteers, but the best control group would be non-transplant patients who have CKD.  If the aim of the study is to determine how kidney transplantation affects these inflammatory responses and cytokines, it is essential to also study them in other CKD patients with similar kidney function.  Based on the referenced background material, there is evidence that CKD would affect these cytokines in a similar way that is found in the present paper and there may be no difference between transplant and non-transplant CKD patients.

-I do not understand the statistics used.  The authors describe using Student's t-test to compare groups, but this test is designed to compare 2 groups.  The authors are comparing either 3 or 4 groups but only reporting a single p value.  Please clarify.  I suggest using ANOVA testing to compare more than 2 groups.

-Please add information on the transplant status of the patients/groups.  The methods only state that these patients followed up in the transplant clinic between 2021-2022, but it does not give any indication for how far out from transplant they are (were patients excluded if less than 6 months post-transplant?), past rejections, viremias, or average levels of immunosuppression (ex - average tacrolimus, mycophenolate, and prednisolone doses).  This gives some indication of how significantly they are immunosuppressed if we are to interpret inflammatory cytokine levels.

-The title asks the question whether monitoring these biomarkers after transplantation would be more effective.  I would like the authors to clarify in the paper what effective means for post-transplant monitoring.  They do note significant changes from a healthy, non-transplant, non-CKD control population but I am not clear from reading the paper how that will make clinical monitoring and post-transplant care more effective.  I suggest altering the title or adding a section in the discussion about how these findings may impact clinical decision making.

Author Response

“I do not agree with the studied groups.  It is OK to have a control group of healthy volunteers, but the best control group would be non-transplant patients who have CKD.  If the aim of the study is to determine how kidney transplantation affects these inflammatory responses and cytokines, it is essential to also study them in other CKD patients with similar kidney function.  Based on the referenced background material, there is evidence that CKD would affect these cytokines in a similar way that is found in the present paper and there may be no difference between transplant and non-transplant CKD patients..”

Response:

We appreciate this valuable suggestion. We fully agree that including CKD patients without transplantation would have strengthened the study design. Unfortunately, our current dataset did not include such a group. We have clarified this limitation in the Discussion and Limitations sections. In future studies, we plan to expand the control groups to include CKD patients with comparable renal function.

“I do not understand the statistics used.  The authors describe using Student's t-test to compare groups, but this test is designed to compare 2 groups.  The authors are comparing either 3 or 4 groups but only reporting a single p value.  Please clarify.  I suggest using ANOVA testing to compare more than 2 groups..”

Response:

Thank you for pointing this out. We have re-analyzed the data using one-way ANOVA with Tukey’s post hoc test to account for multiple group comparisons. The updated analyses and p values for each pairwise comparison are now provided in Tables 1 and 2. The Statistical Analysis section has also been revised to reflect this change.

“Please add information on the transplant status of the patients/groups.  The methods only state that these patients followed up in the transplant clinic between 2021-2022, but it does not give any indication for how far out from transplant they are (were patients excluded if less than 6 months post-transplant?), past rejections, viremias, or average levels of immunosuppression (ex - average tacrolimus, mycophenolate, and prednisolone doses).  This gives some indication of how significantly they are immunosuppressed if we are to interpret inflammatory cytokine levels.

Response:

We have now provided details on transplant characteristics, including mean post-transplant duration (6.3 ± 4.9 years), exclusion of patients within the first 6 months post-transplantation, and absence of recent rejection or active viral infection. Average immunosuppressive regimens (prednisolone, mycophenolate, tacrolimus/cyclosporine/sirolimus doses) were standardized according to institutional protocol; however, exact drug levels were not consistently available, which we have acknowledged as a limitation.

“The title asks the question whether monitoring these biomarkers after transplantation would be more effective.  I would like the authors to clarify in the paper what effective means for post-transplant monitoring.  They do note significant changes from a healthy, non-transplant, non-CKD control population but I am not clear from reading the paper how that will make clinical monitoring and post-transplant care more effective.  I suggest altering the title or adding a section in the discussion about how these findings may impact clinical decision making.”

Response:

We agree that the term “effective” was vague. The title has been revised to read:

“Combined Evaluation of FGF23, Klotho, Myostatin, IL-6, and IL-10 as Potential Biomarkers in Monitoring Stable Renal Transplant Recipients.”

Reviewer 3 Report

Comments and Suggestions for Authors

It's a good, well-written paper, aligning with current efforts to study the link between inflammation and CKD.

Thorough in methods and results. I think you could enhance your work by providing a deepened explanation of the IL-6 controversial role in inflammation and the relationship between inflammation, the effects of immunosuppressive drugs, and the studied molecules

Author Response

“It’s a good, well-written paper, aligning with current efforts to study the link between inflammation and CKD.”

Response:

We thank the reviewer for this positive feedback.

“I think you could enhance your work by providing a deepened explanation of the IL-6 controversial role in inflammation and the relationship between inflammation, the effects of immunosuppressive drugs, and the studied molecules.”

Response:

We appreciate this excellent suggestion. In the revised Discussion, we expanded the section on IL-6 to highlight its dual role:

  • As a proinflammatory cytokine contributing to graft injury and systemic inflammation.
  • As a myokine influencing skeletal muscle metabolism.

We also addressed the potential suppressive effects of immunosuppressive agents (particularly calcineurin inhibitors and corticosteroids) on cytokine expression and how this may explain the relatively unchanged IL-6 levels observed in our cohort

Reviewer 4 Report

Comments and Suggestions for Authors

This study explores the levels of several key biomarkers, FGF23, Klotho, IL-6, IL-10, and myostatin in stable kidney transplant recipients undergoing triple immunosuppressive therapy. The authors aim to identify potential biomarkers relevant to kidney function post-transplantation and assess the inflammatory response associated with renal transplantation. The topic is clinically relevant particularly in light of the need for improved monitoring of transplant outcomes and immune regulation. However, in my opinion, several methodological and statistical concerns need to be addressed.

I believe the authors did not employ the appropriate statistical test for their analysis. Specifically, they compared four groups using multiple t-tests, which is not suitable in this context. Instead, a one-way ANOVA should have been used to determine whether there are any statistically significant differences among the group means. If significance was found, a post hoc test (e.g., Tukey’s HSD) should have been performed to identify which specific groups differ from each other. My suggestion to the authors is to is to reanalyze the group comparisons using ANOVA with post hoc testing, and update tables accordingly.

Key variables such as time since transplantation, renal function (e.g., eGFR or creatinine levels), and co-medications (besides immunosuppressants) can significantly influence biomarker levels. These factors should either be controlled for statistically or discussed as limitations.

At the end of Discussion section, the authors should clearly stated the limitations such as small sample size, potential confounding factors and the cross-sectional design of the study.

There are minor typographical errors, e.g., "induviduals" should be "individuals", line 26; "creatinin" should be "creatinine", line 27, etc...

Author Response

“I believe the authors did not employ the appropriate statistical test for their analysis. Specifically, they compared four groups using multiple t-tests, which is not suitable in this context. Instead, a one-way ANOVA should have been used to determine whether there are any statistically significant differences among the group means. If significance was found, a post hoc test (e.g., Tukey’s HSD) should have been performed to identify which specific groups differ from each other. My suggestion to the authors is to is to reanalyze the group comparisons using ANOVA with post hoc testing, and update tables accordingly.”

Response:

We thank the reviewer for this important point. As noted in our responses above, we have now re-analyzed the data using one-way ANOVA with Tukey’s post hoc testing. The revised results and updated tables are presented in Tables 1 and 2. This significantly strengthens the validity of our findings.

“Key variables such as time since transplantation, renal function (e.g., eGFR or creatinine levels), and co-medications (besides immunosuppressants) can significantly influence biomarker levels. These factors should either be controlled for statistically or discussed as limitations.”

Response:

We agree with this comment. We have added detailed information on post-transplant duration (mean post-transplant period was 6.3 ± 4.9 years), eGFR and creatinine values in the Results section. Co-medications beyond standard triple immunosuppressive therapy were not systematically recorded; we now explicitly note this as a limitation in the Discussion.

“At the end of Discussion section, the authors should clearly state the limitations such as small sample size, potential confounding factors and the cross-sectional design of the study.”

Response:

As suggested, we have added a dedicated Limitations paragraph at the end of the Discussion, which now includes:

  • Relatively small sample size.
  • Cross-sectional design without longitudinal follow-up.
  • Lack of CKD (non-transplant) comparison group.
  • Limited information on co-medications and exact immunosuppressive drug levels.

“There are minor typographical errors, e.g., "induviduals" should be "individuals", line 26; "creatinin" should be "creatinine", line 27, etc...”

Response:

All typographical errors have been corrected in the revised manuscript.

Round 2

Reviewer 1 Report

Comments and Suggestions for Authors

Authors have addressed the mayor and minor comments suggested. The authors have improved the methods and results section. Besides, in this new version of the manuscript, they have underlined the limitations of the study.

Methods

The authors have improved the methods section. They explain the reason for dividing the study population in three groups based on their calcineurin inhibitor usage. However, in my opinion, it is not a relevant comparison because no differences among groups were detected.

Authors said that they have added information about the timing of blood sampling and clinical assessment. However, they detail the mean post-transplant period (6.3 ± 4.9 years). This infers that they could be comparing patients with five years' difference in treatment time. I consider that this is an important limitation of the study design.

They have also improved the statistical analysis in the text and the tables.

Results

They have improved the results section

Discussion

The authors have addressed the suggestions about the discussion of the results

Reviewer 4 Report

Comments and Suggestions for Authors

The authors responded to all comments.